# Reframed GES with a Neural Conditional Dependence Measure

**Xinwei Shen**[1]    **Shengyu Zhu**[2]    **Jiji Zhang**[3]    **Shoubo Hu**[2]    **Zhitang Chen**[2]

[1]Hong Kong University of Science and Technology
[2]Huawei Noah's Ark Lab
[3]Hong Kong Baptist University

## Abstract

In a nonparametric setting, the causal structure is often identifiable only up to Markov equivalence, and for the purpose of causal inference, it is useful to learn a graphical representation of the Markov equivalence class (MEC). In this paper, we revisit the Greedy Equivalence Search (GES) algorithm, which is widely cited as a score-based algorithm for learning the MEC of the underlying causal structure. We observe that in order to make the GES algorithm consistent in a nonparametric setting, it is not necessary to design a scoring metric that evaluates graphs. Instead, it suffices to plug in a consistent estimator of a measure of conditional dependence to guide the search. We therefore present a reframing of the GES algorithm, which is more flexible than the standard score-based version and readily lends itself to the nonparametric setting with a general measure of conditional dependence. In addition, we propose a neural conditional dependence (NCD) measure, which utilizes the expressive power of deep neural networks to characterize conditional independence in a nonparametric manner. We establish the optimality of the reframed GES algorithm under standard assumptions and the consistency of using our NCD estimator to decide conditional independence. Together these results justify the proposed approach. Experimental results demonstrate the effectiveness of our method in causal discovery, as well as the advantages of using our NCD measure over kernel-based measures.

## 1 INTRODUCTION

Causal structure learning is a fundamental problem in various disciplines of science, and flexible solutions to this problem have potentially wide applications [Pearl, 2009, Koller and Friedman, 2009, Peters et al., 2017], e.g., inferring causal relationships among phenotypes [Neto et al., 2010, Zhang et al., 2015], and finding causes in earth system sciences [Runge et al., 2019] and telecommunication networks [Ng et al., 2022]. In many scenarios, it is expensive or even impossible to perform interventions or randomized experiments in order to discover the causal relationships. This limitation inspires the need to infer or at least systematically produce plausible hypotheses of causal structures from purely observational data, which is often known as causal discovery. General assumptions relating the data distribution to the unknown causal structure have been leveraged to make causal discovery feasible, including the well-known Markov condition and faithfulness assumption [Spirtes et al., 2000].

Suppose the unknown causal structure can be properly represented by a directed acyclic graph (DAG) over the observed variables. The last one and a half decades have seen a host of results on the identifiability of the causal DAG from the observational data distribution, under various parametric or semi-parametric assumptions [Shimizu et al., 2006, Hoyer et al., 2009, Zhang and Hyvarinen, 2009, Bühlmann et al., 2014, Peters and Bühlmann, 2014]. However, in a nonparametric setting, from the observational distribution, the causal structure is known to be identifiable only up to Markov equivalence. Despite this limitation, it remains a worthy task to learn a graphical representation of the Markov equivalence class (MEC), known as a completed partial directed acyclic graph (CPDAG), for a CPDAG usually reveals some valuable causal information and can be used to guide experimental studies.

Existing methods for causal discovery targeting the CPDAG are roughly categorized into constraint-based and score-based methods. The former uses statistical tests to find conditional (in)dependence relationships in the data and use them as constraints to recover the CPDAG that satisfies them. The PC algorithm [Spirtes et al., 2000] is a well-known exemplar of this approach. The latter formulates the

*Accepted for the 38th Conference on Uncertainty in Artificial Intelligence* (UAI 2022).

task as an optimization problem by assigning a score to each candidate graph and searching for the one with the optimal score. Regarding the search and optimization strategy, many algorithms solve a combinatorial optimization problem by performing a greedy search; on the other hand, starting from Zheng et al. [2018], much recent work tackles the problem through a continuous optimization [Yu et al., 2019, Lachapelle et al., 2020]. While continuous optimization has advantages in scalability, global convergence is hard to guarantee by using gradient-based algorithms without implausible assumptions such as strong convexity, especially when the model involves neural networks. In contrast, some search algorithms can be shown to achieve global optimality in the large sample limit even with a relatively sparse search space. One of the best-known procedures of this kind is Greedy Equivalence Search (GES) [Chickering, 2002].

The standard score-based GES algorithm requires a scoring criterion to evaluate each candidate graph. Classical examples include the BIC [Schwarz, 1978] and the BDeu scores [Geiger and Heckerman, 1994]. However, most score functions are born out of restrictive parametric assumptions on the data distribution which rarely hold for real-world data. When the parametric model is misspecified, which is very common in real data, the optimality of the standard GES with such a score is not guaranteed to reflect the ground truth even in the large sample limit.

In this paper, we explore a simple strategy to produce a nonparametric GES. We observe that in order to make GES consistent in a nonparametric setting, it is not necessary to design a scoring metric that evaluates graphs as a whole. Instead, it suffices to define a certain criterion to guide the search at each step of the procedure. The approach we consider in this work is to plug in a consistent estimator of a measure of conditional dependence to provide such guidance. The result is a reframed GES algorithm that is more flexible than the standard score-based version and readily lends itself to the nonparametric setting with a general measure of conditional dependence. This avoids potential model misspecification that commonly occurs in score-based methods. On the other hand, although the reframed GES becomes essentially constraint-based, it retains desirable features of the search strategy of GES and performs significantly better in our experiments than paradigmatic constraint-based methods such as PC.

In addition, we propose a measure of conditional dependence based on a characterization of conditional independence from Daudin [1980] and a novel neural conditional dependence (NCD) estimator which utilizes the expressive power of deep neural networks. Many existing nonparametric measures of conditional dependence are based on kernel methods and leverage characterizations in a Reproducing Kernel Hilbert Space (RKHS), e.g., Gretton et al. [2005]. However, kernel methods suffer from high computational complexity, preventing them from efficient applications in

large scale problems. In contrast, our neural network based approach can benefit from a large sample size without a severe compromise in computational time.

We highlight our main contributions as follows:

- We present a reframing of the GES algorithm that can flexibly incorporate a consistent estimator of a general conditional dependence measure.

- We propose a neural conditional dependence (NCD) measure, which utilizes the expressive power of deep neural networks.

- We provide theoretical guarantees on the correctness of the reframed GES algorithm and the consistency of the NCD estimator to measure conditional dependence under mild conditions, and demonstrate the empirical advantages of the resulting method in causal discovery.

## 2 BACKGROUND AND RELATED WORK

### 2.1 PRELIMINARIES AND NOTATIONS

Let $\mathcal{G} = (\mathbf{V}, \mathbf{E})$ be a directed acyclic graph (DAG) consisting of nodes $\mathbf{V} = (X_1, \ldots, X_d)$, each of which is a (possibly multi-dimensional) random variable, and directed edges $\mathbf{E}$ that connect pairs of nodes. Let $\mathbf{Pa}_i^{\mathcal{G}}$ be the set of parents of node $X_i$. We denote the joint distribution of $\mathbf{V}$ by $P_{\mathbf{V}}$. A basic problem of causal discovery aims at inferring the unknown causal DAG $\mathcal{G}$ from an independent and identically distributed (i.i.d.) sample from $P_{\mathbf{V}}$. In general, we need assumptions relating the DAG $\mathcal{G}$ and the distribution $P_{\mathbf{V}}$ to make the task possible. A principle adopted by all causal discovery methods is the *causal Markov condition*: $P_{\mathbf{V}}$ is Markovian with respect to DAG $\mathcal{G}$, in the sense that every conditional independence statement entailed by $\mathcal{G}$ according to the standard Markov property of DAGs is true of $P_{\mathbf{V}}$. We also assume the commonly adopted *faithfulness assumption*: $P_{\mathbf{V}}$ is faithful with respect to DAG $\mathcal{G}$, in the sense that every conditional independence statement true of $P_{\mathbf{V}}$ is entailed by $\mathcal{G}$. If two DAGs $\mathcal{G}_1$ and $\mathcal{G}_2$ entail the same set of conditional independence statements, they are said to be *Markov equivalent*. The set of all DAGs that are Markov equivalent to a DAG $\mathcal{G}$ is called the *Markov equivalence class* (MEC) of $\mathcal{G}$, which can be represented by a completed partially directed acyclic graph (CPDAG). For random variables $X, Y$ and $Z$, we write $X \perp\!\!\!\perp Y \mid Z$ to mean that $X$ and $Y$ are conditionally independent given $Z$.

### 2.2 RELATED WORK ON CAUSAL DISCOVERY

The point that GES can be recast in the spirit of a constraint-based method has been noted in the literature, most recently by Chickering [2020] and most explicitly by Nandy et al. [2018]. To our knowledge, however, the idea of running

GES without a global scoring metric has not been sufficiently explored. In Nandy et al.'s insightful discussion, for example, they emphatically show how a consistent global score can be constructed from local conditional dependence scores in multivariate Gaussian and nonparanormal settings, and stop short of considering the option of dispensing with global scoring altogether. As we aim to demonstrate in this paper, a reframed GES without global scoring is especially flexible and useful in a nonparametric context.

In our experiments, we use a number of state-of-the-art causal discovery algorithms for comparison, in addition to the aforementioned PC and standard GES. One of them is the CAM algorithm [Bühlmann et al., 2014], which decouples the search for the causal ordering from the selection of parents for each variable, by leveraging an additive modeling assumption. Huang et al. [2018] propose a generalized score function (GSF) and apply it in the GES algorithm. Specifically, they transform the statistical decision about conditional independence to a model selection problem for a regression task in an RKHS, define a score based on the penalized log-likelihood for the kernel regression, and then use the score to guide local moves in GES. This work is closely related to ours in that both works are motivated by the goal to develop a nonparametric score to guide the local moves of GES. However, our approach differs from GSF in at least two notable aspects. First, by highlighting the sufficiency of designing a local score that enables consistent statistical decisions about conditional independence, we propose a simpler and more flexible way to dispense with parametric assumptions in GES. Second, the specific score we propose is based on neural networks rather than kernels and hence enjoys better computational efficiency when scaling to large sample size. Another earlier work sharing a similar spirit is the kernel generalized variance (KGV) [Bach and Jordan, 2002], which is also compared in our experiments.

Other methods follow Zheng et al. [2018], who reformulate the original combinatorial problem into a continuous optimization problem, named NOTEARS, which is solved using the augmented Lagrangian algorithm. Several follow-up works extend NOTEARS to nonlinear causal models, including DAG-GNN [Yu et al., 2019], GraN-DAG [Lachapelle et al., 2020], and Ng et al. [2022], all of which utilize neural networks to model the nonlinear causal relations. In addition, Zhu et al. [2020], Wang et al. [2021] adopt policy gradient to search for a DAG with the optimal score.

Since the main purpose of this work is to make GES more applicable in nonparametric settings, in our comparisons we focus mainly on methods that are designed to handle data for continuous variables generated from fairly complex, nonlinear models, and leave out some important methods designed to learn Bayesian networks for discrete variables, such as Bartlett and Cussens [2017].

## 2.3 RELATED WORK ON CONDITIONAL INDEPENDENCE

Conditional independence plays an important role in many statistics and machine learning problems, ranging from graphical models [Koller and Friedman, 2009] to invariance learning [Arjovsky et al., 2019]. A number of studies were devoted to characterizing conditional independence or developing conditional independence tests. Gretton et al. [2005] introduce the Hilbert-Schmidt independence criterion (HSIC), which is extended by Fukumizu et al. [2007] to cover conditional independence and used for a conditional independence test. Recently, Azadkia and Chatterjee [2021] propose a surprisingly simple nonparametric measure of conditional dependence based on ranking statistics, which we refer to as the Rank Conditional Dependence (RCD) measure and will use later to illustrate the flexibility of our approach. Other works focus on constructing tests of conditional independence by proposing various test statistics, including the kernel conditional independence test [Zhang et al., 2011] and the test based on a generalized covariance measure [Shah and Peters, 2020], among others.

# 3 REFRAMING THE GES ALGORITHM

## 3.1 STANDARD GES

The GES algorithm [Chickering, 2002] searches over the space of MECs of DAGs, which are represented by CPDAGs. The connectivity in the search space is given by the independence-map (IMAP) relation: graph $\mathcal{G}$ is an IMAP of graph $\mathcal{G}'$ if every conditional independence entailed by $\mathcal{G}$ is entailed by $\mathcal{G}'$. The standard GES uses a scoring function that assigns a score to every DAG given data, and uses the score of a representative DAG in a MEC as that for the MEC. The search strategy consists of two phases, a phase of forward equivalence search (FES) followed by a phase of backward equivalence search (BES). In FES, the procedure starts with the empty CPDAG (the one with no edges), and moves at each step to a best-scoring CPDAG with one more adjacency (that is an IMAP of the previous CPDAG), until the score cannot be improved by adding more adjacencies. In BES, the procedure starts with the output from FES, and moves at each step to a best-scoring CPDAG with one fewer adjacency (of which the previous CPDAG is an IMAP), until the score cannot be improved by deleting more adjacencies.

We enter some details of FES to highlight the observation that motivates the subsequent reframing. The case of BES is analogous. In FES, each step considers possible insert-one-edge operations on the current CPDAG. Following Chickering [2002], for non-adjacent nodes $X_i$ and $X_j$ in a CPDAG $\mathcal{P}$, and for any subset $\mathbf{T}$ of the neighbors of $X_j$ (i.e., nodes that are connected to $X_j$ by undirected edges) that are not adjacent to $X_i$, the $Insert(X_i, X_j, \mathbf{T})$ operator modifies $\mathcal{P}$

to obtain $\mathcal{P}'$ by inserting the directed edge $X_i \to X_j$, and for each $T \in \mathbf{T}$, directing the previously undirected edge between $T$ and $X_j$ as $T \to X_j$. If the validity condition in Chickering [2002, Theorem 15] is met, then there is a representative DAG $\mathcal{G}$ in the MEC represented by $\mathcal{P}$ and a representative DAG $\mathcal{G}'$ in the MEC represented by $\mathcal{P}'$, such that $\mathcal{G}'$ is the result of inserting $X_i \to X_j$ in $\mathcal{G}$ (which implies that $\mathcal{P}'$ is an IMAP of $\mathcal{P}$).

In Chickering's (2002) proof of the asymptotic correctness of GES under the causal Markov and faithfulness assumptions, the crucial condition is that the "local" decision between $\mathcal{G}$ and $\mathcal{G}'$ mentioned above asymptotically tracks whether a certain conditional independence relation holds. We make this notion precise in the following definition, in which $\mathbf{D}$ denotes an i.i.d. sample with size $n$ from the joint distribution $P_{\mathbf{V}}$ of $\mathbf{V}$.

**Definition 1** (Independence-tracking decision criterion). *Let $\mathcal{G}$ and $\mathcal{G}'$ be two DAGs over $\mathbf{V}$ that are exactly the same except that $\mathcal{G}'$ contains an edge $X_i \to X_j$ that does not appear in $\mathcal{G}$. A decision criterion (based on data $\mathbf{D}$) to choose between $\mathcal{G}$ and $\mathcal{G}'$ (among other options) is independence-tracking if the following two properties hold in the large sample limit:*

*(i) If $X_j \perp\!\!\!\perp X_i \mid \mathbf{Pa}_j^{\mathcal{G}}$ (according to $P_{\mathbf{V}}$), then the decision criterion favors $\mathcal{G}$ over $\mathcal{G}'$.*

*(ii) Otherwise, the decision criterion favors $\mathcal{G}'$ over $\mathcal{G}$.*

In the standard GES algorithm, a scoring function for DAGs is used to make such local decisions. The induced decision criterion is independence-tracking if the scoring function satisfies the so-called local consistency [Chickering, 2002, Definition 6]. Indeed, Definition 1 is a straightforward generalization of the notion of local consistency for scoring functions. The generalization serves to highlight a simple but important observation: the crucial condition for the optimality of GES can be implemented by a (locally consistent) score function for DAGs, but does not necessitate such a function.

## 3.2 REFRAMED GES

We now describe a simple alternative way to implement an independence-tracking decision criterion for GES, by using any consistent measure of conditional dependence, in the following sense:

**Definition 2** ($\tau$-consistency). *Consider a set of statistics $\mathcal{T} = \{T_n(X, Y | \mathbf{Z}) \mid X, Y \in \mathbf{V}, \mathbf{Z} \subseteq \mathbf{V} \backslash \{X, Y\}\}$ (intended to measure conditional dependence) depending on the sample $\mathbf{D}$ with size $n$. $\mathcal{T}$ is said to be $\tau$-consistent with parameter $\tau > 0$ if for every $X, Y \in \mathbf{V}$ and $\mathbf{Z} \subseteq \mathbf{V} \backslash \{X, Y\}$, the following two conditions hold in the large sample limit:*

*(i) If $X \perp\!\!\!\perp Y \mid \mathbf{Z}$ (according to $P_{\mathbf{V}}$), then $T_n(X, Y | \mathbf{Z}) < \tau$.*

*(ii) Otherwise, $T_n(X, Y | \mathbf{Z}) > \tau$.*

For our purpose, the following sufficient condition for the $\tau$-consistency in Definition 2 is useful. All proofs are deferred to Appendix B.

**Proposition 1.** *Suppose for every $X, Y \in \mathbf{V}$ and $\mathbf{Z} \subseteq \mathbf{V} \backslash \{X, Y\}$, $T_*(X, Y | \mathbf{Z}) \geq 0$ is a quantity depending on $P_{\mathbf{V}}$ such that*

$$T_*(X, Y | \mathbf{Z}) = 0 \text{ if and only if } X \perp\!\!\!\perp Y \mid \mathbf{Z}.$$

*Let $\hat{T}_n(X, Y | \mathbf{Z})$ form a set of statistics indexed by $X, Y \in \mathbf{V}$ and $\mathbf{Z} \subseteq \mathbf{V} \backslash \{X, Y\}$. If $\hat{T}_n(X, Y | \mathbf{Z}) \to T_*(X, Y | Z)$ in probability as $n \to \infty$ for every $X, Y \in \mathbf{V}$ and $\mathbf{Z} \subseteq \mathbf{V} \backslash \{X, Y\}$, then there exists $\tau > 0$ such that $\{\hat{T}_n(X, Y | \mathbf{Z})\}$ is $\tau$-consistent.*

This proposition provides a way to construct a $\tau$-consistent set of statistics. One first defines a population quantity that takes the boundary value if and only if the conditional independence in question holds. Then one constructs a consistent estimator for this quantity given an i.i.d. sample. The aforementioned measures including HSIC [Fukumizu et al., 2007] and RCD [Azadkia and Chatterjee, 2021] were both developed along this line. In the next section, we will propose a new measure of conditional dependence based on a neural network implementation.

It is worth noting the essential difference between our defined $\tau$-consistent statistics and conditional independence tests. We note that the condition in Proposition 1 indicates that when $X \perp\!\!\!\perp Y \mid Z$, the statistic $\hat{T}_n(X, Y | \mathbf{Z})$ converges to 0 in probability, i.e., $\hat{T}_n = o_p(1)$. By contrast, in a typical conditional independence test, one usually uses a test statistic that, under the null hypothesis of conditional independence, follows an asymptotic null distribution, which is then used to develop a decision rule. This means that when $X \perp\!\!\!\perp Y \mid Z$, the test statistic is stochastically bounded, i.e., $O_p(1)$, but not necessarily $o_p(1)$. Therefore, it is in general non-trivial to define a $\tau$-consistent statistic from a conditional independence test.

With a $\tau$-consistent $\hat{T}(X, Y | \mathbf{Z})$, it is straightforward to implement an independence-tracking decision criterion to be used in GES. Specifically, at each step in FES, to each (valid) operator $Insert(X_i, X_j, \mathbf{T})$ we assign $\hat{T}(X_i, X_j | \mathbf{Pa}_j^{\mathcal{G}})$ as its "local score" (where $\mathcal{G}$ is the DAG representing the current CPDAG induced by the operator), and apply the operator with the highest local score (indicating conditional dependence), unless all remaining valid insert operators yield a local score lower than the threshold $\tau$. Similarly, at each step in BES, to each (valid) operator $Delete(X_i, X_j, \mathbf{H})$, we assign $\hat{T}(X_i, X_j | \mathbf{Pa}_j^{\mathcal{G}})$ as its local score (where $\mathcal{G}$ is the DAG representing the CPDAG the operator would produce),

**Algorithm 1** The update step in the reframed FES
***
**Input**: the current CPDAG $\mathcal{P}$, sample $\mathbf{D}$, a list of valid insert operators **INS**, statistics $\hat{T}(X, Y|\mathbf{Z})$, threshold $\tau$
**Output**: the next CPDAG $\mathcal{P}'$

    Set $s = 0$ and $I = \text{NULL}$.
    **for** $Insert(X_i, X_j, \mathbf{T}) \in$ **INS do**
        Let $\mathcal{G}$ be the representative DAG of $\mathcal{P}$ corresponding to $Insert(X_i, X_j, \mathbf{T})$.
        Evaluate $Score(X_i, X_j, \mathbf{T}) = \hat{T}(X_i, X_j | \mathbf{Pa}_j^{\mathcal{G}})$.
        **if** $Score(X_i, X_j, \mathbf{T}) > s$ **then**
            Let $s = Score(X_i, X_j, \mathbf{T})$ and $I = Insert(X_i, X_j, \mathbf{T})$.
        **end if**
    **end for**
    **if** $s > \tau$ **then**
        Apply operator $I$ to obtain $\mathcal{P}'$.
    **else**
        Keep $\mathcal{P}' = \mathcal{P}$ (and terminate FES).
    **end if**
    **return** $\mathcal{P}'$
***

and apply the operator with the lowest local score (indicating conditional independence), unless all remaining valid delete operators yield a score greater than $\tau$. The update step of FES is summarized in Algorithm 1, and the dual update step of BES is given in Appendix A.

We call the GES algorithm with these update steps the *reframed GES*. Unlike the standard GES, this reframed GES does not optimize a global score for MECs. However, by using a suitable local score for choosing operators to apply (or to stop), the local decision criterion remains independence-tracking, and as a result the asymptotic optimality of the reframed GES algorithm is still guaranteed, as stated in the following theorem.

**Theorem 2.** *Under the causal Markov and faithfulness assumptions, the reframed GES procedure using a $\tau$-consistent $\hat{T}(X, Y|\mathbf{Z})$ recovers the MEC of the true graph in the large sample limit.*

## 4 NEURAL CONDITIONAL DEPENDENCE MEASURE

In this section, we propose a novel measure of conditional dependence. Let $X$, $Y$, and $Z$ be three random variables taking values in $\mathbb{R}^{d_X}$, $\mathbb{R}^{d_Y}$, and $\mathbb{R}^{d_Z}$, respectively, where $d_X$, $d_Y$, and $d_Z$ are the corresponding dimensions. We assume that their joint distribution is absolutely continuous with respect to Lebesgue measure with density $p_*$ defined on $\mathbb{R}^{d_X + d_Y + d_Z}$. The conditional independence between $X$ and $Y$ given $Z$ is defined by $p_*(x, y, z) = p_*(x|z)p_*(y|z)p_*(z)$ for all $x, y, z$ with $p_*(z) > 0$ [Dawid, 1979].

The following lemma from Daudin [1980] characterizes the conditional independence, which has given rise to several hypothesis testing methods. Let $L_Z^2$, $L_{XZ}^2$, and $L_{YZ}^2$ be the

spaces of square integrable functions of $Z$, $(X, Z)$, and $(Y, Z)$, respectively, e.g., $L_{XZ}^2 = \{f : \mathbb{R}^{d_X + d_Z} \to \mathbb{R} \mid \mathbb{E}[f(X, Z)^2] < \infty\}$.

**Lemma 3** (Daudin [1980]). *The random variables $X$ and $Y$ are conditionally independent given $Z$ if and only if*

$$\mathbb{E}[f(X, Z)g(Y, Z)] = 0,$$

*for all $f \in L_{XZ}^2$ and $g \in L_{YZ}^2$ such that $\mathbb{E}[f(X, Z)|Z] = 0$ and $\mathbb{E}[g(X, Z)|Z] = 0$.*

At the population level, given a ground truth density $p_*$, we propose the following measure of conditional dependence between $X$ and $Y$ given $Z$:

$$S(X, Y|Z) = \sup_{f,g} \rho^2(f(X, Z) - h^*(Z), g(Y, Z) - l^*(Z)) \quad (1)$$

where $f \in L_{XZ}^2$ and $g \in L_{YZ}^2$ are test functions, $h^*(Z) = \mathbb{E}[f(X, Z)|Z]$, $l^*(Z) = \mathbb{E}[g(Y, Z)|Z]$, and $\rho(X_1, X_2) = \text{cov}(X_1, X_2)/\sqrt{\text{var}(X_1)\text{var}(X_2)}$ denotes the Pearson correlation coefficient of two random variables $X_1$ and $X_2$. The reason for using the correlation coefficient rather than the covariance is that after normalization by the variances, the characteristic is bounded between $[-1, 1]$. This makes the measure well-defined in a bounded range and the computation of its subsequent sample version numerically stable.

Based on Lemma 3, we have the following simple theorem which characterizes the property of the measure (1) and establishes an equivalence condition between the measure and conditional independence.

**Theorem 4.** *For all $p_*$, we have $S(X, Y|Z) \in [0, 1]$ and $S(X, Y|Z) = 0$ if and only if $X \perp\!\!\!\perp Y \mid Z$.*

Having defined the measure $S(X, Y|Z)$, we now make the computation tractable. We use deep neural network classes to parametrize the test functions $f, g$ and the conditional expectations $h, l$ in (1). Formally, we write $f_\theta$, $g_\phi$, $h_\omega$, $l_\psi$, where the subscripts denote the parameters of the corresponding neural networks. We then exploit the approximation

$$\sup_{\theta, \phi} \rho^2(f_\theta(X, Z) - h_{\omega^*}(Z), g_\phi(Y, Z) - l_{\psi^*}(Z)), \quad (2)$$

where $h_{\omega^*}(z) = h^*(z)$ and $l_{\psi^*}(z) = l^*(z)$. According to the universal approximation theorem of neural networks [Hornik et al., 1989], equation (2) can approximate the true measure (1) with arbitrary accuracy by choosing the appropriate network architecture. Since here we mainly focus on the statistical property of the estimator proposed below, we ignore the small approximation error (i.e., the gap between (2) and (1)) in the analysis for simplicity.

Next, we present a consistent estimator of $S(X, Y|Z)$. Let $\mathbf{D} = \{(x_i, y_i, z_i), i = 1, \ldots, n\}$ be the collection of i.i.d.

**Algorithm 2** Computing the NCD score

**Input**: sample $\mathbf{D}$, horizon $T_t, T_r$, initial $\theta, \phi, \omega, \psi$
**Output**: NCD score
1: **for** $t_t = 1, 2, \ldots, T_t$ **do**
2:   **for** $t_r = 1, 2, \ldots, T_r$ **do**
3:     Update $\omega$ by descending $\sum_i \nabla_\omega (f_\theta(x_i, z_i) - h_\omega(z_i))^2$
4:     Update $\psi$ by descending $\sum_i \nabla_\psi (g_\phi(y_i, z_i) - l_\psi(z_i))^2$
5:   **end for**
6:   Update $\theta, \phi$ by ascending the gradient of
      $\hat{\rho}^2(f_\theta(X, Z) - h_\omega(Z), g_\phi(Y, Z) - l_\psi(Z))$
7: **end for**
8: Compute $\hat{s} = \hat{\rho}^2(f(X, Z) - h(Z), g(Y, Z) - l(Z))$
9: **return** $\hat{s}$

---

copies of $(X, Y, Z) \sim p_*$. Our estimator of $S(X, Y|Z)$ is given by

$$\hat{S}_n(X, Y|Z) = \sup_{\theta, \phi} \hat{\rho}^2 \big( f_\theta(X, Z) - h_{\hat{\omega}}(Z), g_\phi(Y, Z) - l_{\hat{\psi}}(Z) \big), \quad (3)$$

where $\hat{\rho}$ is the sample correlation coefficient based on data $\mathbf{D}$, and

$$\hat{\omega} = \arg\min_\omega \frac{1}{n} \sum_{i=1}^n (f_\theta(x_i, z_i) - h_\omega(z_i))^2,$$
$$\hat{\psi} = \arg\min_\psi \frac{1}{n} \sum_{i=1}^n (g_\phi(y_i, z_i) - l_\psi(z_i))^2, \quad (4)$$

are the estimators of $\omega^*$ and $\psi^*$.

*Remark.* The estimators in (4) based on regression come from the fact that $\mathbb{E}[f(X, Z)|Z] = \arg\min_h \mathbb{E}[f(X, Z) - h(Z)]^2$, which is proved in Appendix B.

We call the proposed estimator (3) the *neural conditional dependence (NCD)* estimator and its population version (2) the NCD measure. Since (3) solves a bilevel optimization problem involving regression problems (4), we adopt an alternating gradient descent scheme to obtain the NCD estimator. The procedure is summarized in Algorithm 2, where we alternately update the test functions $f_\theta, g_\phi$ and nonlinear regressors $h_\omega, l_\psi$ for $T_t$ and $T_r$ steps, respectively.

To study the asymptotic behavior of the proposed estimator, we assume the following regularity conditions, all of which are mild assumptions commonly adopted in the literature.

*C1.* The parameter spaces $\theta \in \Theta$, $\phi \in \Phi$, $\omega \in \Omega$, and $\psi \in \Psi$ are compact.

*C2.* $f_\theta(x, z)$, $g_\phi(y, z)$, $h_\omega(z)$, and $l_\psi(z)$ are continuous with respect to the corresponding parameters and data $x, y, z$.

*C3.* $f_\theta(x, z)$, $g_\phi(y, z)$, $h_\omega(z)$, and $l_\psi(z)$ are dominated square integrable, i.e., there exists a dominating function $F(x, z)$ such that $|f_\theta(x, z)| \leq F(x, z)$ for all $\theta$ and $\mathbb{E}[F(X, Z)]^2 < \infty$.

*C4.* For all $\theta, \phi$, there exist unique $\omega^*(\theta) \in \Omega$ and $\psi^*(\phi) \in \Psi$ such that $h_{\omega^*}(z) = h^*(z)$ and $l_{\psi^*}(z) = l^*(z)$ almost surely, respectively.

The following theorem establishes the consistency of $\hat{S}_n(X, Y|Z)$ as an estimator of the population measure $S(X, Y|Z)$.

**Theorem 5.** *Under the regularity conditions C1-C4, as $n \to \infty$, we have $\hat{S}_n(X, Y|Z) \to S(X, Y|Z)$ in probability.*

Finally, we apply the NCD measure to causal discovery through the reframing of GES in the previous section. We plug the estimator $\hat{S}_n$ into the reframed GES procedure as the "local score". Based on Theorems 4 and 5, by applying the sufficient condition in Proposition 1, we know that $\hat{S}_n(X, Y|\mathbf{Z})$ satisfies the $\tau$-consistency in Definition 2 with some $\tau > 0$. Then Theorem 2 implies the asymptotic correctness of our method to recover the true MEC.

In addition, to demonstrate the flexibility of our reframed GES algorithm in incorporating various conditional dependence measures, we will also test a version of our procedure using the RCD measure recently proposed by Azadkia and Chatterjee [2021], because it is very easy to compute. The RCD estimator can be shown to satisfy the $\tau$-consistency and hence suits our framework well. For completeness, we provide a description of RCD in Appendix C.

## 5 EXPERIMENTS

In this section, we compare our proposed method with various existing state-of-the-art causal discovery approaches on both synthetic and pseudo-real data sets. Baseline methods include score-based methods using BIC [Chickering, 2002], KGV [Bach and Jordan, 2002], and GSF [Huang et al., 2018]; a constraint-based method, PC algorithm [Spirtes et al., 2000]; a method based on structural causal model, CAM [Bühlmann et al., 2014]; as well as the emerging methods in the continuous optimization paradigm including NOTEARS [Zheng et al., 2018], DAG-GNN [Yu et al., 2019], and GraN-DAG [Lachapelle et al., 2020]. The details of the experimental settings and hyperparameters (including the choice of $\tau$) of baseline methods and ours are given in Appendix D.[1]

The causal discovery performance is evaluated using three metrics: the structural hamming distance (SHD), the structural interventional distance (SID) [Peters and Bühlmann, 2015] and the F1 score. Since our method and many baseline approaches return a CPDAG representing an MEC, both SHD and SID are evaluated between the learned and ground-truth CPDAGs. Then the SHD is the smallest number of

---

[1] Our code is available at https://github.com/xwshen51/GES-NCD.

| Setting | GP (1k) | | GP (5k) | | MULT (1k) | | MULT (5k) | |
|---|---|---|---|---|---|---|---|---|
| Methods | SHD | F1 score | SHD | F1 score | SHD | F1 score | SHD | F1 score |
| NCD | **5.6±2.5** | **0.63±0.14** | **4.2±2.3** | **0.71±0.14** | 6.2±2.9 | 0.59±0.08 | 5.6±2.4 | 0.60±0.08 |
| RCD | 9.0±0.7 | 0.41±0.07 | 8.4±1.1 | 0.53±0.08 | 7.4±2.1 | 0.51±0.09 | **3.2±1.3** | **0.67±0.07** |
| PC | 8.8±1.6 | 0.36±0.15 | 7.2±2.4 | 0.50±0.16 | 7.6±1.7 | 0.44±0.15 | 4.6±1.8 | 0.57±0.13 |
| BIC | 7.0±2.8 | 0.49±0.20 | 6.0±2.5 | 0.59±0.17 | 4.2±2.9 | 0.65±0.09 | 4.4±3.4 | 0.62±0.11 |
| KGV | 8.5±1.1 | 0.37±0.08 | 7.5±0.5 | 0.51±0.06 | 9.0±1.9 | 0.35±0.14 | 7.2±0.7 | 0.47±0.07 |
| CAM | 6.0±3.5 | 0.50±0.26 | 7.2±3.7 | 0.52±0.22 | 10.8±1.8 | 0.09±0.07 | 11.2±2.3 | 0.13±0.15 |
| NOTEARS | 11.4±0.9 | 0.06±0.08 | 11.6±0.9 | 0.06±0.08 | 24.8±3.8 | 0.36±0.07 | 23.6±4.7 | 0.37±0.07 |
| DAG-GNN | 11.0±1.7 | 0.00±0.00 | 11.4±1.8 | 0.03±0.07 | 16.4±2.6 | 0.37±0.13 | 13.6±3.4 | 0.40±0.10 |
| GraN-DAG | 10.6±1.1 | 0.05±0.06 | 12.2±1.8 | 0.12±0.04 | 8.6±2.6 | 0.54±0.12 | 10.2±1.9 | 0.51±0.08 |
| GSF | 6.4±3.5 | 0.55±0.19 | >12h | | **3.0±1.1** | **0.67±0.06** | >12h | |

Table 1: SHD and F1 score on PNL data sets with 10 nodes, 2 expected degrees, and 1000 and 5000 samples.

| Setting | GP (1k) | | GP (5k) | | MULT (1k) | | MULT (5k) | |
|---|---|---|---|---|---|---|---|---|
| Methods | SHD | F1 score | SHD | F1 score | SHD | F1 score | SHD | F1 score |
| NCD | **28.4±3.6** | **0.55±0.05** | **24.6±3.8** | **0.58±0.08** | **29.2±4.6** | **0.52±0.07** | 29.8±5.1 | **0.57±0.09** |
| RCD | 32.8±2.2 | 0.39±0.11 | 32.6±4.7 | 0.44±0.10 | 31.4±4.5 | 0.43±0.14 | **27.2±3.3** | 0.54±0.06 |
| PC | 37.6±1.3 | 0.18±0.09 | 36.0±2.7 | 0.26±0.05 | 36.2±1.8 | 0.23±0.06 | 34.4±1.5 | 0.32±0.08 |
| BIC | 33.0±2.1 | 0.45±0.06 | 30.8±3.3 | 0.50±0.09 | 30.8±5.6 | 0.43±0.09 | 35.0±3.9 | 0.39±0.07 |
| KGV | 37.8±0.7 | 0.20±0.08 | 34.2±3.9 | 0.33±0.07 | 37.2±1.6 | 0.27±0.02 | 37.4±2.2 | 0.31±0.06 |
| CAM | 33.0±5.6 | 0.42±0.13 | 30.6±3.4 | 0.50±0.11 | 35.2±2.8 | 0.25±0.07 | 34.4±6.5 | 0.31±0.15 |
| NOTEARS | 38.8±1.9 | 0.13±0.05 | 38.4±1.8 | 0.13±0.05 | 39.0±1.6 | 0.33±0.04 | 39.0±1.9 | 0.34±0.07 |
| DAG-GNN | 39.2±1.3 | 0.03±0.02 | 39.2±2.3 | 0.05±0.09 | 37.8±2.4 | 0.26±0.10 | 39.6±1.1 | 0.25±0.12 |
| GraN-DAG | 34.0±7.9 | 0.18±0.09 | 35.4±6.9 | 0.30±0.13 | 37.4±3.2 | 0.20±0.08 | 37.0±3.5 | 0.27±0.09 |
| GSF | 34.0±3.0 | 0.39±0.05 | >12h | | 31.6±3.2 | 0.38±0.09 | >12h | |

Table 2: SHD and F1 score on PNL data sets with 10 nodes, 8 expected degrees, and 1000 and 5000 samples.

edge additions, deletions, and reversals to convert the estimated CPDAG into the true CPDAG. The SID counts the number of pairs $(X_i, X_j)$ such that the interventional distribution $p(x_j|do(X_i = x))$ would be miscalculated if we chose the parent adjustment set from the estimated graph. We report the SIDs corresponding to the best and worst DAG in the learned MEC. The SHD and SID are computed using functions corresponding to CPDAGs in the Causal Discovery Toolbox [Kalainathan et al., 2020]. The F1 score is defined as the harmonic mean of the precision and the recall. Computing the F1 score involves summarizing the number of correctly estimated edges. Directed edges in the ground-truth CPDAG are deemed correctly estimated if the learned CPDAG contains exactly the same directed edge and are deemed incorrectly otherwise. Undirected edges in the ground-truth CPDAG are converted to two directed edges in the adjacency matrix. When the learned CPDAG contains exactly the same undirected edge, both converted directed edges are correctly estimated. One directed edge and no edge in the learned CPDAG are deemed as correctly estimating 1 and 0 edge, respectively. In general, a lower SHD or SID and a higher F1 score indicate a better estimate.

## 5.1 SYNTHETIC DATA

As mentioned in previous sections, when a data set does not satisfy the additive Gaussian noise assumption, many existing methods such as BIC, CAM, NOTEARS, and GraN-DAG suffer from model misspecification and thus may lead to misleading results. In contrast, nonparametric methods like GSF and ours in principle will not be affected. Here we consider the well-known post nonlinear (PNL) causal models [Zhang and Hyvarinen, 2009]. A general PNL model expresses each variable $X_i$ as

$$X_i = g_{i,2}(g_{i,1}(\mathbf{Pa}_i) + N_i), \ i = 1, \ldots, d,$$

where $\mathbf{Pa}_i$ contains the direct causes of $X_i$, $N_i$ is the exogenous noise variable, and $g_{i,1}$ and $g_{i,2}$ are nonlinear transformations.

To synthesize a data set, we first randomly generate a ground-truth DAG $\mathcal{G}$ following the Erdős-Rényi (ER) graph model and then generate data following $\mathcal{G}$ and two types of PNL models that were also considered in Lachapelle et al. [2020]. The first one, called *PNL-GP*, samples $g_{i,1}$ independently from a Gaussian process with bandwidth one, takes $g_{i,2}$

as the sigmoid function, and $N_i \sim Laplace(0, b_i)$ with $b_i \sim \mathcal{U}[0, 1]$. All root variables in PNL-GP are sampled from $\mathcal{U}[-1, 1]$. The second one, named *PNL-MULT*, takes $g_{i,1}(x) = \log(sum(x))$ where $sum(x)$ takes the sum of all components of a vector $x$, $g_{i,2}(\cdot) = \exp(\cdot)$, and $N_i \sim |\mathcal{N}(0, \sigma_i^2)|$ with $\sigma_i^2 \sim \mathcal{U}[0, 1]$. All root variables in PNL-MULT are sampled from $\mathcal{U}[0, 2]$. This model is adapted from Zhang et al. [2015].

Tables 1 and 2 present the results of SHD and F1 score on sparse and dense graphs with 10 nodes respectively, where the error bars represent the standard deviations across 5 data sets per setting. The results of SID are basically consistent, which are deferred to Appendix E due to the space limit. Additional results on graphs with 20 nodes are also presented in Appendix E. We see that in general, the reframed GES algorithm with our own NCD or the adopted RCD (shown in the first two lines of all tables) performs the best across all settings, except in the sparse PNL-MULT data where GSF is the best. The advantages of our methods on the more challenging dense graphs are more significant than those on sparse ones. In most cases, NCD outperforms RCD, though RCD produces excellent results on PNL-MULT with a larger sample. From the perspective of implementation, RCD may be favored over NCD in terms of fewer hyperparameters and less computational cost. In addition, our methods exhibit similarly good performances across different ground-truth models, while most other methods tend to perform well on at most one setting, which indicates the robustness of our nonparametric approach against different distributions.

GSF, as another kernel-based nonparametric score, performs very well on PNL-MULT with a sparse structure, but is less competitive in other settings. Note that we only report the results of GSF using 1000 samples, because even for the sparse graph, it takes around 17 hours for a single run with 5000 samples compared to around 19 minutes with 1000 samples. In contrast, our NCD-based method can benefit from a larger sample size while taking similar computational time as with a smaller sample (both within 4 minutes in the sparse case). In Appendix E, we discuss more details regarding the computational time of different methods. KGV leads to inferior performance in all settings. The standard GES with the linear-Gaussian BIC score sometimes performs well on PNL-MULT; a possible reason is that when the noise variance $\sigma_i^2 \sim \mathcal{U}[0, 1]$ happens to be small, the PNL-MULT model behaves similarly to a linear-Gaussian model, leading to a case with minor misspecification for BIC. This may also partly account for the fact that PC performs better on PNL-MULT than on PNL-GP; that is, a PNL-MULT data set can be similar to linear-Gaussian data which would satisfy the model assumption made in the hypothesis testing. CAM performs better on PNL-GP than on PNL-MULT and achieves the best SID in one case, as shown in Appendix E. The continuous optimization methods are inferior on these PNL data sets, which could be explained

| Method | SHD | SID | F1 score |
|---|---|---|---|
| NCD | **2.6±2.9** | **[2.0±3.7, 14.2±2.8]** | **0.73±0.15** |
| PC | 3.0±1.3 | [7.4±4.3, 14.9±3.7] | 0.57±0.11 |
| KGV | 4.1±1.6 | [3.7±3.3, 17.8±2.8] | 0.58±0.11 |
| DAG-GNN | 4.3±2.3 | [4.1±3.9, 15.6±4.1] | 0.59±0.27 |
| GSF | 4.7±3.0 | [4.3±4.2, 15.6±3.0] | 0.61±0.17 |
| BIC | 4.7±0.9 | [4.8±3.8, 16.2±2.7] | 0.57±0.00 |

Table 3: Results on 10 multi-dimensional data sets.

by their misspecification of the model.

In addition, we evaluate our methods in a multi-dimensional scenario where each node may have more than one dimension. Note that our NCD estimator can be readily applied to the multi-dimensional setup by adjusting the input dimension of the test functions, while the rank-based RCD measure unfortunately cannot be directly applied here. Some of the baseline methods, including CAM, NOTEARS, and GraN-DAG do not apply to the multi-dimensional case, so we do not compare with them in this setting. We use 10 synthetic data sets from Huang et al. [2018], each with 5 nodes and a sample size of 1000. As shown in Table 3, our approach outperforms the baseline methods in all three metrics. PC, KGV, and GSF are the second-best performing methods in terms of SHD, SID, and F1 score, respectively, though they all give an inferior performance in other metrics.

## 5.2 PSEUDO-REAL DATA

Although the synthetic data sets from PNL models can expose the model misspecification problem in many existing methods, they differ from the additive noise setup only by the nonlinearity $g_{i,2}$, and hence amount to relatively mild cases of misspecification. In this section, we consider a pseudo-real data set sampled from the SynTReN generator [Van den Bulcke et al., 2006] where there is no guarantee at all for model specification. We evaluate on the 10 data sets sampled by Lachapelle et al. [2020], each with 20 nodes and a small sample size of 500. In addition, we consider a real Bayesian network, CHILD network (with 20 nodes), and randomly generate 3000 samples following the PNL-GP model introduced in the previous section.

As shown in Table 4, on SynTReN, most baseline methods perform poorly, indicating a potentially severe violation of their model assumptions. Our reframed GES with NCD and with RCD obtain the best SHDs. The results on SynTReN suggest the potential advantage of nonparametric causal discovery methods in real applications where model misspecification is common and possibly grave. Moreover, our methods also obtain the best performance on the real graph structure CHILD. Note that on this large data set with 3000 samples, the kernel-based methods GSF and KGV face serious computational challenges in that they take more than 12

| Method | SynTReN | CHILD |
|--------|---------|-------|
| NCD | **30.0±5.8** | **16.8±3.3** |
| RCD | **30.9±4.8** | **14.0±5.8** |
| PC | 37.4±4.1 | 22.6±9.4 |
| BIC | 65.8±10.8 | 32.8±16.8 |
| NOTEARS | 99.8±14.4 | 23.6±1.9 |
| DAG-GNN | 38.5±5.1 | 28.8±3.2 |
| GraN-DAG | 58.7±10.0 | 17.2±2.1 |
| KGV | 39.9±8.0 | >12h |
| GSF | 52.1±8.9 | >12h |

Table 4: SHD on pseudo-real data.

hours for a single run. Therefore, we expect our method to exhibit even more advantages than the kernel-based methods in large-scale scenarios.

## 6 CONCLUSION

In this work, we presented a reframed GES algorithm that works with a measure of conditional dependence rather than a scoring metric for graphs. This way the algorithm is easily applicable in a nonparametric setting with a theoretical guarantee. We also proposed a neural conditional dependence (NCD) measure based on a deep neural network implementation, and established its theoretical properties that make it suitable for the reframed GES. The resulting causal discovery algorithm was shown in our experiments to be superior or competitive in comparison to a number of state-of-the-art methods. It also enjoys a significant advantage over kernel-based nonparametric methods in large-scale settings, since the latter are usually infeasible when the sample size is relatively large. For future work, we plan to explore a continuous optimization formulation of causal discovery based on such nonparametric conditional dependence measures.

**Acknowledgements**

JZ's research was supported in part by the RGC of Hong Kong under GRF13602720 and a start-up fund from HKBU. The authors thank the anonymous reviewers for their valuable comments and suggestions.

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
