# OpenReview forum: "Reframed GES with a Neural Conditional Dependence Measure"
_auai.org/UAI/2022/Conference — UAI 2022 Poster_

### Official Review · Reviewer_twbA · 2022-04-11

**Q2(1) Originality/Novelty:** 3
**Q2(2) Significance/Impact:** 3
**Q2(3) Correctness/Technical Quality:** 3
**Q2(6) Clarity Of Writing:** 3
**Q6 Overall Score:** 5
**Q8 Confidence In Your Score:** 4

**Q1 Summary And Contributions:**

The authors use a version of GES driving by conditional independence testing, along with a conditional independence test implemented using deep neural networks, to analyze fairly generally distributed simulated data. The experimental results look to be quite good, compared to a number of competitors.

**Q2 Assessment Of The Paper:**

More detailed information regarding each of these aspects is given below:

**Q2(4) Quality Of Experiments (Optional):**

4: Excellent: The experimental evaluation is comprehensive and the results are compelling.

**Q2(5) Reproducibility:**

4: Excellent: Key resources (e.g., proofs, code, data) are available and key details (e.g., proof sketches, experimental setup) are comprehensively described for competent researchers to confidently and easily reproduce the main results.

**Q3 Main Strengths:**

The subject matter taken up in this paper is very helpful, as methods for analyzing the type of data used in the experimental section are still in need. The specific models used for testing are well-selected, and the analyses given for them is compelling. The choice of algorithms for comparison makes sense. Also, the method given here shows promise for analyzing some difficult data.

**Q4 Main Weakness:**

One thing given up by going to an independence-based GES, it seems, is the ability a score-based GES has to sort through all scores at each step to find the edge addition/deletion that would yield the optimal such score, so on the fact of it, an independence-based GES is somewhat inferior to a score-based GES. Perhaps this is not true for the implementation in this paper, but the issue should probably be discussed somewhere; if it was, I missed it. In addition, timing results would have been very much appreciated. Also, it would be helpful to clarify why a score-based approach wasn't pursued in the first place, or in addition.

**Q5 Detailed Comments To The Authors:**

The FGES algorithm cited is not the original FGES algorithm; perhaps a reference to the original would be helpful in addition.

Tuning the network to achieve a sparsity given by expert knowledge is less helpful than tuning, say, to minimize reversals of edges whose orientations are given by expert knowledge.

The writing is a bit hard to follow in places; perhaps another editing pass would help.

**Q7 Justification For Your Score:**

A discussion of why an independence-based GES should be able to compete with a score-based GES would be helpful. I feel that such a discussion could be added to the paper, improving it.

**Q9 Complying With Reviewing Instructions:**

1: Yes.

---

### Official Review · Reviewer_uqri · 2022-04-11

**Q2(1) Originality/Novelty:** 3
**Q2(2) Significance/Impact:** 3
**Q2(3) Correctness/Technical Quality:** 3
**Q2(6) Clarity Of Writing:** 3
**Q6 Overall Score:** 7
**Q8 Confidence In Your Score:** 3

**Q1 Summary And Contributions:**

The authors propose a modification of Greedy Equivalence Search (GES) algorithm, which they call reframed GES. It is based on tau-consistent statistics. They also propose a novel conditional independence measure - a neural conditional dependence measure - and use it in the modified GES algorithm.

**Q2 Assessment Of The Paper:**

More detailed information regarding each of these aspects is given below:

**Q2(4) Quality Of Experiments (Optional):**

2: Fair: The experimental evaluation is weak: important baselines are missing, or the results do not adequately support the main claims.

**Q2(5) Reproducibility:**

2: Fair: Key resources (e.g., proofs, code, data) are unavailable but key details (e.g., proof sketches, experimental setup) are sufficiently well-described for an expert to confidently reproduce the main results.

**Q3 Main Strengths:**

The idea of using a tau-consistent statistics as a CI-measure guiding the GES algorithm is interesting. The experimental results of the GES algorithm with the proposed CI measure are convincing.

**Q4 Main Weakness:**

I miss comparisons with methods that guarantee optimal solutions and that are based on other principles than greedy equivalence search, as is for example the Gobnilp method [1]. I also miss more detailed comparisons of the computational time, computational time is mentioned several times in the paper in a non-systematic way. I wonder how computationally demanding is the actual learning of the neural network based estimator and how it affects the overall computational complexity of the GES algorithm.

References
[1] https://www.cs.york.ac.uk/aig/sw/gobnilp/

**Q5 Detailed Comments To The Authors:**

Page 1
Why the claim that causal structure learning has potentially wide applications is not supported by references to applications but by references to books on foundations of causal models and probabilistic graphical models?
In Introduction several different methods for structural learning are listed. However, an important recent method [2] from the family of score-based methods is not mentioned.
Definition 1
Symbol n is not defined.
Page 5, Section 4
Symbols d_X, d_Y, and d_Z are not defined.
In the definition of conditional independence of X and Y given Z the term p(z) on RHS or conditioning on Z in the term on LHS is missing.
Please, unify references, e.g. David Maxwell Chickering and Max Chickering is the same person and should be referred in the same way. Either initials or full first names should be used consistently. This citation is awfully confused:  Judea Pearl et al. Models, reasoning and inference. Cambridge, UK: Cambridge University Press, 19, 2000.

References
[2] M. Bartlett and J. Cussens. Integer linear programming for the Bayesian network structure learning problem. Artificial Intelligence, 244:258 – 271, 2017. ISSN 0004-3702. https://doi.org/10.1016/j.artint.2015.03.003.


**Q7 Justification For Your Score:**

Intersting idea with a theoretical justification. Convincing experiments. Computational time is not reported in a systematic way.

**Q9 Complying With Reviewing Instructions:**

1: Yes.

---

### Official Review · Reviewer_9sPt · 2022-04-12

**Q2(1) Originality/Novelty:** 3
**Q2(2) Significance/Impact:** 3
**Q2(3) Correctness/Technical Quality:** 3
**Q2(6) Clarity Of Writing:** 3
**Q6 Overall Score:** 7
**Q8 Confidence In Your Score:** 2

**Q1 Summary And Contributions:**

This paper reframes the standard GES algorithm for a general measure of conditional dependence. The authors made the interesting observation that the consistency of standard GES relied on a local decision criterion and reframed GES to only rely on such a criterion. Using a consistent measure of conditional dependence as such local decision criterion shows that asymptotic optimality of reframed GES can be guaranteed. To this end, they propose the neural dependent dependence measure.

**Q2 Assessment Of The Paper:**

More detailed information regarding each of these aspects is given below:

**Q2(4) Quality Of Experiments (Optional):**

3: Good: The experimental evaluation is adequate, and the results convincingly support the main claims.

**Q2(5) Reproducibility:**

3: Good: Key resources (e.g., proofs, code, data) are available and key details (e.g., proofs, experimental setup) are sufficiently well-described for competent researchers to confidently reproduce the main results.

**Q3 Main Strengths:**

Overall, the core content of the paper is well written and organized. The author's observation that the consistency of GES is based on a local decision criterion, that asymptotically tracks whether a conditional independence relation holds, and can be replaced by any consistent measure of conditional (in)dependence is a significant and novel insight. They relate conditional independencies to a \tau-consistent test statistic and show asymptotic optimality is guaranteed for these test statistics. This allows for the incorporation of various conditional dependence measures, such as their novel NCD measure, and this makes their algorithm more broadly applicable. Moreover, they clearly discuss the limits and implications of their work.



**Q4 Main Weakness:**

Compared to the rest of the paper, the readability of the abstract and introduction can be improved. For example, it doesn't say anything about the local decision criterion, which I think is one of the main insights of the paper. It would be beneficial for the paper to state the significant insight and contributions clearly in the abstract. The introduction could be improved (see also my comments below). For example, the second paragraph on page 2 comes a bit out of the blue. This paragraph is trying to make the point that parametric models have their problems. But how does this relate to GES and to reframed GES?

The fact that the proofs of the main theoretical results are contained in the supplementary material makes this paper less self-contained. Although this is the only slight weakness of the paper; however, I did not check all the proofs there, to be honest.


**Q5 Detailed Comments To The Authors:**

Minor comments:
- p.1 - abstract: "... justify ..." <-- To me, it is not immediately clear why these results justify the approach?
- p.1: "... in order to reveal the causal ..." <-- "reveal" or "discover"?
- p.1: "... seen a host of nice results" <-- The word "nice" is a bit subjective
- p.2: "... continuous program" <-- What program?
- p.2: "... some search algorithms" <-- Is there a citation missing?
- p.2: "... born out of ..." <-- I don't know what the authors mean here
- p.2: "nonparametric GES" <-- Is GES parametric?
- p.2: "The way we consider in this work ..." <-- There may be something missing here
- p.2: "... paradigmatic constraint-based" <-- What is a paradigmatic constraint-based method?
- p.4 - Def. 1: Do we assume that $X_i\subsetneq \p_j^G$?
- p.4 - Def. 2: What does the subscript n mean? Does (i) and (ii) hold for all n?
- p.4: "One first defines a population quantity ..." <-- This sentence is difficult to understand because of the "if and only if" in the sentence
- p.5 - Algorithm 1: What does "valid" mean?
- p.5: "... numerically stable." <-- Why is this? Or a reference is missing?


**Q7 Justification For Your Score:**

In my opinion, this paper provides an interesting and novel approach to causal discovery. I think this work can be a valuable contribution to UAI, under the condition that all the proofs are correct (which I didn't check) and that the overall storyline in the abstract and introduction is improved.


**Q9 Complying With Reviewing Instructions:**

1: Yes.

---

### Official Review · Reviewer_4NqC · 2022-04-13

**Q2(1) Originality/Novelty:** 3
**Q2(2) Significance/Impact:** 3
**Q2(3) Correctness/Technical Quality:** 3
**Q2(6) Clarity Of Writing:** 3
**Q6 Overall Score:** 7
**Q8 Confidence In Your Score:** 3

**Q1 Summary And Contributions:**

This paper reformulates GES using a local measure of conditional dependence rather than a scoring criterion. While this has previously been discussed in the literature, it has not been formalized and the authors make such a procedure rigorous. They proceed to introduce an estimate of conditional dependence using neural networks and also apply a recent measure of dependence using ranking statistics. Both of these methods are highly competitive in simulation and scales to larger sample sizes.

**Q2 Assessment Of The Paper:**

More detailed information regarding each of these aspects is given below:

**Q2(4) Quality Of Experiments (Optional):**

3: Good: The experimental evaluation is adequate, and the results convincingly support the main claims.

**Q2(5) Reproducibility:**

2: Fair: Key resources (e.g., proofs, code, data) are unavailable but key details (e.g., proof sketches, experimental setup) are sufficiently well-described for an expert to confidently reproduce the main results.

**Q3 Main Strengths:**

The main strength of this paper is the introduction of an algorithm that achieves state-of-the-art performance for causal DAG learning with few nodes described by a PNL model. Moreover, the algorithm is able to run in a fraction of the time of its strongest competitor (a RKHS-based score).

**Q4 Main Weakness:**

The NCD measure of conditional dependence requires setting up and training a neural network. This means that a practitioner who might want to use GES with NCD is required to set many hyper-parameters. I might have missed it, but I did not see a discussion on how one might choose/fit those parameters.

**Q5 Detailed Comments To The Authors:**

I suggest moving the contents of Appendix D2 to the main text. When I was reading the results in the main text, I did not know how to inperpret the F1-score.

I suggest including a table of run times for the methods run in the various simulated scenarios.

The choice of hyper-parameters for NCD is included in the appendix, but how were these parameters chosen? Do you have any recommendation for how someone else using your method should choose these parameters? I suggestion adding such a discussion to the paper.

**Q7 Justification For Your Score:**

The paper is well written, clear, and impactful. However, I would like to see more of a discussion on recommendations for using the method in practice.

**Q9 Complying With Reviewing Instructions:**

1: Yes.

---

### Decision · Program_Chairs · 2022-05-15

**Decision:**

Accept (Poster)

**Comment:**

Meta Review: The paper proposes a GES algorithm with a local measure of conditional dependence instead of a global score commonly used previously.

Pros:

-Local instead of global is potentially impactful.

-The major benefit is that they can handle complex non-linear relationships.

-The method is justified by theoretical results.

Cons:

-Some unclear statements and references missing, which the authors intend to correct.

The recommendation is an accept.